# SlideAugment: A Simple Data Processing Method to Enhance Human Activity Recognition Accuracy Based on WiFi

**DOI:** 10.3390/s21062181

**Published:** 2021-03-20

**Authors:** Junyan Li, Kang Yin, Chengpei Tang

**Affiliations:** School of Intelligent Systems Engineering, Sun Yat-sen University, Guangzhou 510006, China; lijy395@mail2.sysu.edu.cn (J.L.); 245403188gm@gmail.com (K.Y.)

**Keywords:** Wi-Fi, channel state information, data augmentation, slide window, human activity recognition

## Abstract

Currently, there are various works presented in the literature regarding the activity recognition based on WiFi. We observe that existing public data sets do not have enough data. In this work, we present a data augmentation method called window slicing. By slicing the original data, we get multiple samples for one raw datum. As a result, the size of the data set can be increased. On the basis of the experiments performed on a public data set and our collected data set, we observe that the proposed method assists in improving the results. It is notable that, on the public data set, the activity recognition accuracy improves from 88.13% to 97.12%. Similarly, the recognition accuracy is also improved for the data set collected in this work. Although the proposed method is simple, it effectively enhances the recognition accuracy. It is a general channel state information (CSI) data augmentation method. In addition, the proposed method demonstrates good interpretability.

## 1. Introduction

Human activity recognition forms the basis of human–computer interaction. The common solutions for activity recognition are usually based on computer vision [1], sensors [2], and wearable devices [3]. Vision is an important way for humans to obtain information of the surrounding environment. However, the activity recognition based on vision is strongly influenced by many factors, such as light conditions, line of sight, and background information. These limitations seriously affect the output of the computer vision algorithms used in various applications [4]. Similarly, the information can also be collected through sensors and wearable devices. The wearable devices include smart watches, smart bracelets, and smart phones. These devices are equipped with sensors and require users to wear them all the time in order to collect information [3,5]. Therefore, it is essential to investigate the recognition technology that is insensitive to users and does not involve user privacy. WiFi-based sensing, as a contactless sensing technology, has attracted increasing attention in recent years. In addition, its low-cost and easy-to-deploy features are also one of the reasons for its popularity.

Before 2013, the received signal strength indication (RSSI) from WiFi signals was widely used in wireless sensing. A study presented by Tsinghua University shows that in complex dynamic environment, the channel state information (CSI) from WiFi performs better than RSSI [6] in the field of indoor location. Since then, the channel state information from physical layer has been explored in activity recognition [7], indoor location [8], gesture recognition [9,10], and user identification [11]. The WiFi signals can be used to perform fall detection [12] and detect risky driving behavior [13]. Many studies focus on how to process CSI data and most common solutions are based on machine learning algorithms, such as support vector machine(SVM) [14] and convolutional neural network(CNN) [15]. There are also some special solutions presented considering the characteristics of CSI, such as attention based bi-directional long short-term memory (ABLSTM) [16] and multiple kernel representation learning (MKRL) framework [17]. In addition to studying which algorithm to use, studying how to apply WiFi based activity recognition to other fields is also a new and meaningful research direction. Currently, campus violence is a common phenomenon. However, it is difficult to obtain relevant evidence. The researchers have proposed WiVi, a ubiquitous violence detection system with commercial WiFi devices [18]. A special contactless and fine-grained gesture recognition method using CSI, namely Wi-SL, is proposed to assist people suffering from aphasia and hearing impairment to effectively interact with the computer and facilitate their daily life [19].

A data set is a sampling of real data. In the field of statistical machine learning, the quality of the data set determines the quality of the data analysis result. We collect the existing data sets in the field of WiFi based activity recognition (this research has been submitted to the Proceedings of the ACM on Interactive, Mobile, Wearable and Ubiquitous Technologies (IMWUT) journal in November 2020) and observe that the existing data sets suffer from a common issue, i.e., the amount of data is insufficient. The largest available data set is the Widar3.0 data set published by Tsinghua University [20]. The Widar3.0 data set is divided into two parts. The first part contains 12,000 gesture samples and the second part holds 5000 instances. In 2020, a data set was published on the Journal Data in Brief [21]. Compared with previous data sets, which solely focus on the interaction between the human and the computer, this data set focuses on the interaction between people. Each action is completed by a pair of volunteers. There are 4800 instances (40 pairs × 12 interactions × 10 trials) in this data set. In a nutshell, the existing data sets generally comprise several thousand records, which is significantly smaller than the data sets used in natural language processing and computer vision. At the moment, the most commonly used solutions are based on deep learning. So, the deep learning algorithms cannot be trained efficiently. Thus, the models tend to memorize the training samples and overfit.

In computer vision field, the techniques, such as data augmentation, geometric transformations, color space augmentations, kernel filters, mixing images, random erasing, are very useful to enhance the size and quality of the training data sets [22]. The effectiveness of these methods depends on the immutability of image’s semantic information. For example, even though a picture of a cat is stretched and rotated, it can still convey the same semantic information: there is a cat in the picture. However, when the information represented by CSI data is flipped or rotated, it changes in an unpredictable fashion. There are many similarities between the CSI information and time series data. So, we try to learn from some data augmentation methods for time series data. However, there are few studies focus on augmentation for time series data. The authors in [23] present a data augmentation method called window slicing for enhancing the small scale data sets. The authors apply the window slicing on various time series data sets to prove the effectiveness of this method. Therefore, we attempt to apply the window slicing method to the CSI data. We obtain multiple samples after applying the window slicing on a single sample of CSI data. By adjusting the window size and slicing rules, we can get multiple samples for a single CSI data. So, the size of training data set usually multiplies. 

In order to prove the effectiveness of this method, we perform experiments on the basis of ARIL [24], a public data set. ARIL provides the CSI data and two corresponding labels, one indicates the activity type and the other indicates the location. The training data (1116 pieces) and test data (276 pieces) have been divided. We download the data and code from website, and train the CSI fingerprints with activity labels to obtain an activity recognition model. Then, we test this model on the given test set to get a baseline. The window slicing method is applied on train data to get multiple samples. After that, we use the new samples to train the activity recognition model. Then, the model is evaluated on the same test set. The experimental results show that the recognition accuracy of the model is significantly improved after using the window slicing method, from 88.13% to 97.12%. Additionally, we also collect the data from two indoor environments and construct a data set named CSIGe. We perform the same experiments on CSIGe as the experiments performed on ARIL. The experiments on both ARIL and CSIGe show that the window slicing method has the ability to significantly enhance the accuracy of human activity recognition. The three major contributions of this work are as follows.

We carefully analyze the existing data sets and methods proposed for WiFi based human activity recognition and identify the issues. We observe that the public data sets generally have insufficient data and the records are pre-processed insufficiently.Considering the existing problems, we first present a window slicing method for CSI data. By slicing on the original data, the size of data set increases. We perform experiments on the existing data sets to verify the effectiveness of the proposed method.We present a WiFi based activity recognition data set named CSIGe. Furthermore, we perform experiments on CSIGe to verify the effectiveness of the proposed window slicing method.

The rest of the manuscript is organized as follows: 

In Section 2, we present the background knowledge of RSSI and CSI. The published data sets for WiFi based human activity recognition are also introduced in this section. In Section 3, we present the proposed method. In Section 4, we present the experiments. In Section 5, we discuss the results. In Section 6, we conclude this work.

## 2. Preliminary

### 2.1. RSSI and CSI 

RSSI and CSI are two measurements for WiFi signals. Specifically, RSSI indicates the received signal strength (RSS) for the signal received from the media access control (MAC) layer, which is affected by path attenuation, occlusion and multipath effect. CSI, which reflects the channel state variations on the sub-channel level, is a physical layer (PHY) feature of the received signal. Compared with RSSI, CSI contains higher granular information of WiFi channels, which includes signal strength, phase variation, and frequency shift. However, the processing on CSI is more complicated and computationally unfriendly than RSSI.

The RSSI represents the received power in decibels (dB), which is mathematically defined as
(1)$$\mathrm{RSSI}=10\log_{2}{\left\|V\right\|}^{2}$$
where *V* denotes the signal voltage [25]. The authors present a work that focuses on the indoor location based on RSSI of the wireless local area network (WLAN) interfaces [26]. It is notable that in a complex environment, the performance of RSSI is unstable. The authors in [27] present a novel calibration approach for RSSI. In 2013, a work presented by the Tsinghua University showed that the CSI (reflecting the channel response in 802.11 a/g/n) is able to discriminate multipath characteristics, and thus holds the potential for the convergence of accurate and pervasive indoor localization [6]. 

The multipath effect of wireless channel is generally described by the channel impulse response (CIR). Under the assumption of linear time invariance, the channel impulse response is mathematically expressed as
(2)$$H(k)=\left\|H(k)\right\|e^{j∠H(k)}$$
where *H*(*k*) denotes the CSI information for the *k*-th subcarrier, the ‖H(K)‖ represents the amplitude for the *k*-th subcarrier, and the ∠H(k) denotes the phase for the *k*-th subcarrier.

In the field of wireless communication, CSI is the channel attribute of a communication link. It describes the attenuation factor of the signal in each transmission path, i.e., the value of each element in the channel gain matrix H (sometimes referred to as channel matrix or channel fading matrix), such as signal scattering, multipath fading, shadowing fading, and power decay of distance, etc. The CSI allows the communication system to adapt the current channel conditions and guarantee the high-reliability and high-rate communication in multi antenna systems.

In brief, RSSI is mainly used for indoor location and tracking tasks [28]. Compared to RSSI, CSI is fine-grained physical information and is more sensitive to environment. Therefore, it is usually used for activity recognition [29], gesture recognition [30], keystroke recognition [31], and tracking [32]. 

### 2.2. The Survey of Public Data Sets for WiFi Based Activity Recognition

In previous work (which was submitted to the The Proceedings of the ACM on Interactive, Mobile, Wearable and Ubiquitous Technologies (IMWUT) journal in November 2020), we collect data sets published recently and the results are shown in Table 1. Wih2h is the latest data set published in Data in Brief [18], which concentrates on human-to-human actions. Most notably, two volunteers work together to complete the assigned actions when collecting the experimental data. This work also focuses on the experimental setup and data acquisition, and it does not provide any baseline based this data set. For the WiAR data set [21], the data are organized by the actors. In the original work, the activity types, the data format, the ways of data acquisition. Then, influencing factors are explained [21]. In order to evaluate the performance of WiAR data set on the activity recognition, five classification algorithms and two deep learning algorithms are applied. The Widar3.0 is the largest data set in WiFi sensing area [27]. The raw data is stored according to the date of collection and there is a file named “README.pdf”, which describes the collected data in detail. According to the description in the original work, the data set is divided into two parts. The first data set contains 12,000 gesture samples (16 users × 5 positions × 5 orientations × 6 gestures × 5 instances). The second data set is composed of 5000 samples (2 users × 5 positions × 5 orientations × 10 gestures × 10 instances) [27]. In order to solve the problem of action recognition and indoor location simultaneously, ARIL [24] was proposed. The CSI fingerprints are collected by two universal software radio peripherals (USRPs). They define 6 hand gestures, namely, hand up, hand down, hand left, hand right, hand circle and hand cross. For one CSI data, there are two labels, one indicates the location and the other indicates the gesture type. Although the data is collected in only one room, the volunteer stands at 16 different positions and performs the assigned actions. In the original paper, they introduce a novel 1-dimensinal CNN to achieve the joint task of indoor localization and activity recognition named ResNet1D. The data and the code are publicly available at https://github.com/geekfeiw/ARIL (accessed on 19 July 2019). The Signfi data set contains most types of gestures, including up to 276 sign language gestures [10]. 

## 3. Methods

In this section, we present the data augmentation method and the experiment settings used for data collection.

### 3.1. Window Slicing

Inspired by time series classification [29], we believe that for existing CSI data sets, the data augmentation can also be used to increase the amount of data and enhance the recognition accuracy. After reading the experimental setup of data collection in many research, we discover a common phenomenon. When collecting the CSI data, a piece of data generally includes two pieces of blank information (i.e., it does not contain any action information, only static environment information). Such as WiAR data set, the author pointed out that each activity sample was collected with more than 7 s data which contain 2–3 s activity data (effective data) and 4–6 s empty data (indoor environment data) [25]. We use this window slicing method based on this premise. By sliding the window on the original data, we get multiple samples for the same action, which can be understood as recording the same action at different times and getting multiple records of one action. The labels of new samples are same as the original data. A CSI subcarrier is presented in Figure 1. We take a piece of data from ARIL as an example to illustrate the widow slicing method. The length of one subcarrier in raw data is 192. We set the window size as 180, so after window slicing, we get 13 samples for one piece of data. 

We apply window slicing on CSI data, which is similar to the time series data. For one subcarrier in CSI data *C* = {C_t1_, C_t2_, ..., C_tn_}, a slice is a snippet of the subcarrier in original CSI data defined as *S_i:i+j_* = {C_ti_, C_ti + 1_, ..., C_ti + j_}, 1 ≤ i ≤ n, 0 ≤ j < n − i. The i(ti) marks the beginning of the data and the j(tj) equals to the window size. 

The existing data set named ARIL provides the data and the source code [24], which is useful for reproducing experiments. Therefore, we conduct experiments on this data set to compare the effects of the original experiment (without data augmentation) and the new experiments (with data augmentation). On a given training data set, we applied the window slicing method. So, one original training data generates multiple samples, which are considered as independent instances. When predicting the labels, we do not apply the window slicing on test data set. 

### 3.2. Data Collection

#### 3.2.1. Hardware

In order to measure CSI, we use two TPLINK N750 routers (two Atheros AR9580 Wi-Fi chipsets respectively loaded on the two industrial computers). One is transmitter (Tx) and the other is the receiver (Rx). The receiver router is equipped with three antennas with an interval of 0.1m, while the transmitter router is equipped with only one antenna. The bandwidth is 40 MHz, and the Wi-Fi frequency is 5 GHz. So, there are 114 subcarriers for each Tx-Rx pair. The sampling rate is 1000 packet/s.

#### 3.2.2. Environment Settings

We collect data in two indoor environments. In the first environment, we marked three positions on which the volunteers stand and perform designated actions. In the second environment, two locations are marked. As presented in Figure 2, the interval between the transmitter and the receiver is 2.6 m and the volunteer stands in the designated position facing the acquisition device. The volunteer sees the instructions on a screen placed in front of him/her. The details of the two indoor environments are shown as follows.

The Room1 is a 10 × 13 m room. There are three desks with the size of 5 × 2 m in Room1. There are three windows on the right side of the room. The transmitter(Tx) and receiver(Rx) are separated by 2.6 m. The Room2 is a 10 × 4 m room. There are two desks in Room2, the size of the desk on the left is 3 × 1 m and the size of right desk is 7 × 2 m. There is a window on the right side of the room. The distance between transmitter(Tx) and receiver(Rx) is 2.6 m.

The five volunteers stand at five different positions (three in room1 and two in room2) and complete the six designated gestures. Each gesture is repeated 20 times. So, we collected 3000 (5 volunteers × 5 positions × 6 gestures × 20 instances) instances. Each instance has four labels, which include the identity of volunteers, room number, location number, and action category.

#### 3.2.3. Activity Types

We collect six common gestures, which are similar to the gestures in ARIL. These gestures include hand left, hand right, hand up, hand down, hand circle, and draw zigzag (from Widar3.0). These six gestures are presented in Figure 3. Please note that these gestures are commonly used in the field of human-computer interaction. The first four gestures have a well-defined directionality. In contrast, the last two gestures are more complex. We use an automated collection platform to collect the CSI information containing the specified gesture. The platform is developed by a member of our team. When collecting data, we place a monitor opposite to the volunteer to display the action acquisition instructions. A server is placed behind the monitor to provide the computing services and monitor the operations of the system. The volunteers stand at the designed location and face the monitor. Before data collection, we need to present the action category and collection times. After the operations of the system starts, the display requests the volunteer to raise the hand upwards. Please note that the preparation time is 3 s. After 3S, the volunteers start performing the action, and the system starts to collect data packets. The duration of data collection is 1.8 s. Afterwards, the display stops for 2S, and the volunteers can have a short break. Please note that the preparation time and rest time after collection can be adjusted.

## 4. Experiments

### 4.1. Window Slicing on ARIL

In order to verify the effectiveness of the proposed window slicing method, we reproduce the experiment in the paper according to the data and source code presented in ARIL [24]. In the original paper, the researchers proposed a novel deep learning framework ResNet1D for joint activity recognition and indoor localization task using WiFi channel state information (CSI) fingerprints. ResNet1D is based on ResNets, a famous and classic network structure [33]. Since ResNets were proposed, it has been widely used in image classification [33], video classification [34] and speech emotion recognition [35]. The standard ResNets are designed to process the images, a 2-dimensional data with height and width. Therefore, the researchers made some modifications to its structure in order to process one-dimensional data. The detailed introduction of ResNet1D can be found in Figures 10 and 11 of the original paper [24] or on the website https://github.com/geekfeiw/ARIL (accessed on 19 July 2019). When training the novel framework, it’s input is WiFi CSI fingerprints and two labels, one indicates the location, the other indicates the type of activity. According to the experiments, they achieved 88.13% accuracy for activity recognition and 95.68% for indoor location when using the novel framework ResNet1D-[1,1,1,1]. They achieved 89.57% accuracy for activity recognition and 95.68% for indoor location when using the novel framework ResNet1D-[1,1,1,1]+. Data and code have been made publicly available at https://github.com/geekfeiw/ARIL (accessed on 19 July 2019). We download the data and code from this link. In order to focus on the activity recognition task, we modify the framework by removing the indoor location recognition brunch. So, when reproducing the experiments in ARIL, we only take the WiFi CSI fingerprints and activity type label as input. The experiment results show that, we achieve 88.13% accuracy for activity recognition when using the novel framework ResNet1D-[1,1,1,1]. For the ResNet1D-[1,1,1,1]+ framework, we achieve 88.49% accuracy for activity recognition. In terms of training parameters, we use the settings similar to the settings of the original work. The network is trained for 150 epochs by Adam opitimizer with default settings (β1 = 0.9, β2 = 0.999). We set the mini-batch size as 128 and the initial learning rate is 0.005. The learning rate decays by 0.5 every 10 epochs [24]. We implement the experiments on PyTorch-1.8.0, framework on an Intel(R) Xeon(R) CPU E5-2630 v4 @2.20GHz, with an Nvidia Titan X Pascal GPU and 32.0 GB of RAM. 

For action recognition task, when training ResNet-1D [1,1,1,1] model, the accuracy is around 88.13%, while when training the ResNet-1D [1,1,1,1]+ model, the test accuracy is around 88.49%. Please note that the format of raw training data is [1116,52,192], where 1116 denotes the number of training samples, 52 represents the number of subcarriers, and 192 represents the size of each CSI fingerprint stream. On this basis, we add the window slicing algorithm during the training phase. We select different window sizes, get new sampling data, and carry out the experiments on new data. When testing the model, we use the raw test data without performing data augmentation. The format of raw test data is [2766,52,192], where 276 represents the number of test samples, 52 represents the number of subcarriers and 192 represents the size of each CSI fingerprint stream. From the Figure 4, we notice that the training loss curve keeps relatively steady after the 50th epoch in original experiment. After using window slicing method, the train loss curve stabilizes before 50th epoch (Figure 5). In addition, the test curve reaches closer to zero than the test curve in the original experiment. The experimental results presented in Table 2 show that the recognition accuracy of the two models is significantly improved. 

We perform additional experiments to verify the effectiveness of the slicing window algorithm. First, we use the original length of 192 training samples to get the trained model and save it. Then, we use the window slicing method on raw train data to get multiple samples. Finally, we continue to train the model on new data. Please note that the test set still uses the original test set without any processing. For example, the window size is set to 180. When slicing the time dimension of the original data, we get 13 samples for one piece data. The experimental results presented in Table 3 show that the recognition accuracy of the two models is significantly improved when the window size is reduced.

For the two aforementioned experiments, the accuracy of the test set increases significantly. As discussed in the orignial paper [24], the volunteers keep standing and not perform any action at the beginning and end of data collection. So, there are two blank stages in each datum. Therefore, when we slice on the raw data, the small part of data that is removed contains almost no effective action information. It means that although the size of the data has changed, the amount of information contained in the data remains unchanged. Furthermore, the slicing of the original data is equivalent to the sampling at different times. In other words, when collecting data, we use multiple groups of devices to record CSI information at different times. Finally, we combine all the records to get a group of action data. Therefore, when performing tests on test data, no matter how long the blank information before and after the action in the test set is included, the accuracy is not affected as such situations are already learned by the model.

### 4.2. Data Collection and Processing

In data acquisition experiment, we plan to collect 3000 samples of actions and record the video of the action with the camera. The video information can be used as an auxiliary information in the later data preprocessing. Additionally, we can also study the multimodal task of video information combined with CSI information, similar to the work presented in CVPR 2019 [36]. Therefore, when collecting data, the gesture type, the number of repetitions, the preparation time, the acquisition time, and the rest time are set in advance. After starting data acquisition, there is only one volunteer performing the actions in the environment. The display provides the instructions and the volunteers complete the specified action in the specified time according to the instructions. The video is also recorded. After data collection, we preprocess and save the data.

We assess the validity of the acquired data in two aspects. First, we check if there is an invalid information in the obtained CSI information. Second, we watch the action video to ensure there are no wrong actions and other unwanted characteristics. After data processing, we get 2844 valid data samples, i.e., 1702 from room1 and 1142 from room2. The numbers of gestures collected by each volunteer is presented in Table 4. 

### 4.3. Window Slicing on CSIGe

We also perform experiments on the proposed CSIGe dataset using window slicing algorithm. We conduct experiments on the data collected from room1 and room2, respectively. The format of data in CSIGe in room1 is [1702,3,114,1800]. The first dimension, i.e., 1702 represents the number of samples, the second dimension, i.e., 3 represents equivalent to the number of antenna pairs, 114 represents the number of subcarriers, and 1800 (1.8 s × 1000 packet/s) represents the data with a length of 1.8 s. In order to align the data format with the format of ARIL, we adjust the data format in room1 to [1702,342,1800]. Because of the large scale of CSI information, it is not convenient for our equipment to train the model using the new data obtained after the slicing operation. Therefore, we only conduct the second experiment on CSIGe for ARIL. We set the window size as 1790. First, we train the raw data and test on the test data set and save the trained model. Then, we slice the original training data to get multiple samples. Finally, we train the saved model using the new data. It is important to note that the raw data are split to train and test set by 4:1. We implement the experiments on PyTorch-1.8.0, framework on an Intel(R) Xeon(R) CPU E5-2630 v4 @2.20GHz, with an Nvidia Titan X Pascal GPU and 32.0 GB of RAM. The initial learning rate is 0.0001 which decays by 0.5 after every 10 epochs. The network is trained for 200 epochs. Table 5 shows that when we train the saved model on new samples, the recognition accuracy is improved.

## 5. Discussion

We focus on WiFi based activity recognition in this research. First of all, we summarized the existing research in this field, and found that it can be divided into two directions, one is the basic research for methodological research, and the other is the applied research for expanding the application fields. The current popular research methods are usually based on machine learning algorithm, and the performance of the algorithm largely depends on the data set used. So, we focus on data sets. Through the investigation of the existing public data sets, we find that the common problem is insufficient data. Thus, we propose a data augmentation method called window slicing. This method is based on the characteristics of data collection. To improve the effectiveness of this method, we performed experiments on a public data set ARIL and our data set CSIGe. The results show that window slicing method enhances the recognition accuracy. 

In our experiments, we set different sizes of the window to get multiple samples for raw data. The results show that window slicing method enhances the recognition accuracy. However, we do not pay attention to how big the window size is, which improves the recognition accuracy the most. Actually, when the window size becomes smaller, the number of samples for each category increases, and the size of data set increases. However, it should be noted that the amount of information contained in the new sampling results is decreasing. Therefore, it is very difficult to define the most suitable window size. We believe that the effectiveness of this method has been preliminarily confirmed in the research, which provides a new idea for related researches. In addition to solving the problem of insufficient data, we believe that this method should also have potential in solving the problem of imbalanced data. In short, this is a simple but effective data augmentation method on CSI data.

## 6. Conclusions

In this work, we propose a simple data augmentation method to enhance the human gesture recognition accuracy based on WiFi. The proposed window slicing method is inspired by time series data classification tasks and multi-scale sampling data enhancement methods. In order to confirm the effectiveness of the proposed method, we conduct experiments on a public data set named ARIL [24] and our data set CSIGe. Moreover, we design and develop a CSI information self-service collection system. The results show that window slicing method can increase the amount of data and then enhance the recognition accuracy. This is a general data augmentation method for CSI information. In forthcoming work, we will conduct further research on the collected data set CSIGe.

## Figures and Tables

**Figure 1 sensors-21-02181-f001:**
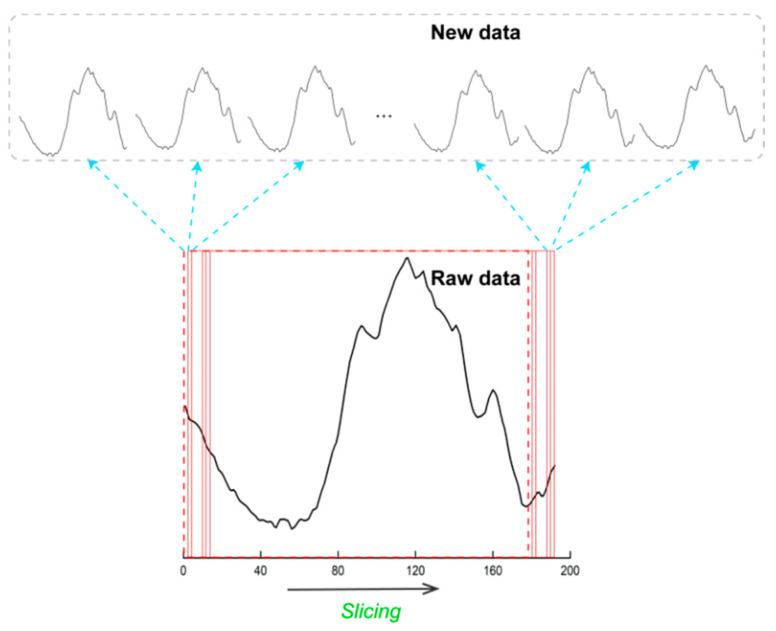
An example to explain the window slicing algorithm.

**Figure 2 sensors-21-02181-f002:**
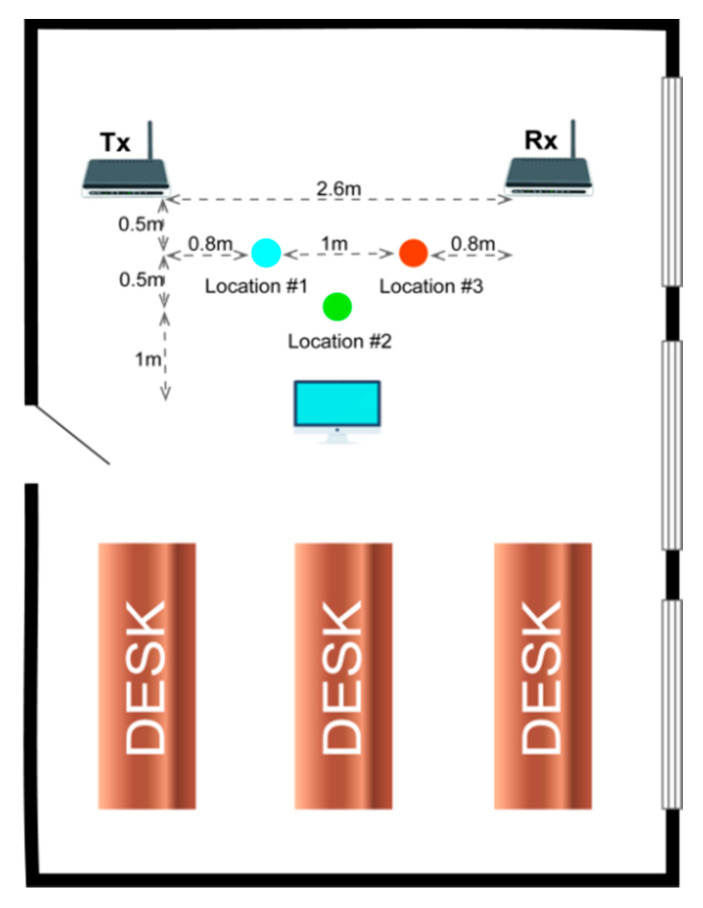
The layout of the two indoor environments, the upper one is Room1 and the lower is Room2.

**Figure 3 sensors-21-02181-f003:**
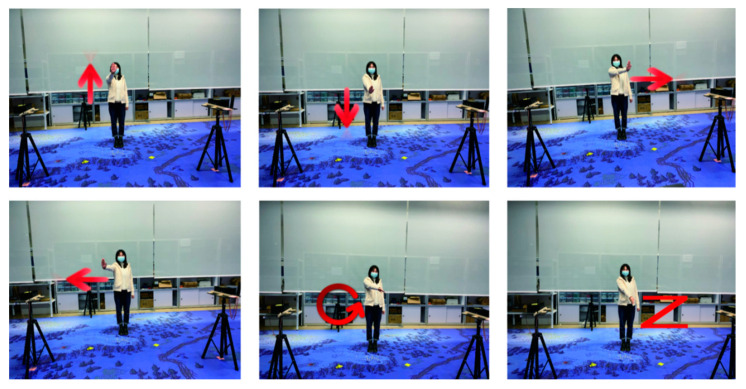
The six common gestures, including hand up, hand down, hand left, hand right, hand circle, and draw zigzag.

**Figure 4 sensors-21-02181-f004:**
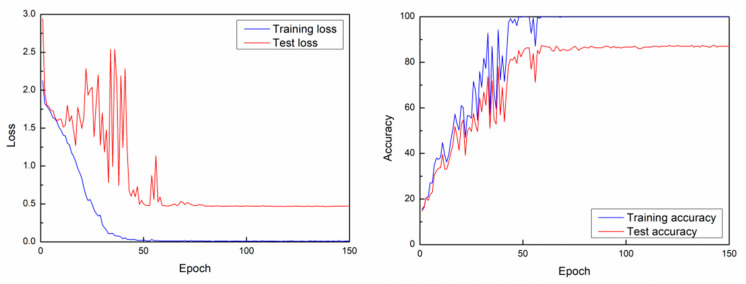
Learning curves and losses of training raw data (the size of training data is [1116,52,192]). The network is ResNet1D-[1,1,1,1].

**Figure 5 sensors-21-02181-f005:**
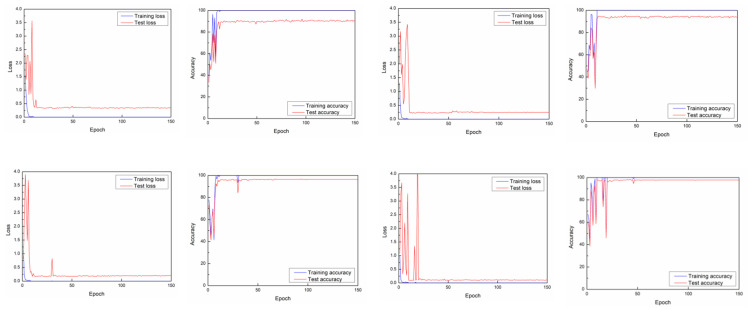
Learning curves and losses of window slicing method when the window size is 180, 170, 160 and 150. The network model is ResNet1D-[1,1,1,1].

**Table 1 sensors-21-02181-t001:** The summary of the public data sets of WiFi-based human activity recognition.

Data Set Name	Year	Activity Types	No. of Actors	No. of Scenarios	Total Instances
Wih2h [21]	2020	12	40 pairs	1	4800
WiAR [25]	2019	16	10	3	4800
Widar3.0 [20]	2019	16	16	3	17,000
ARIL [24]	2019	6	1	1	1394
Signfi [10]	2018	276	5	2	8280

**Table 2 sensors-21-02181-t002:** The accuracy of action recognition when the size of slice window is 192, 180, 170, 160, 150.

Window Size	192	180	170	160	150
ResNet1D-[1,1,1,1]	88.13%	90.29%	92.45%	96.76%	97.12%
ResNet1D-[1,1,1,1]+	88.49%	90.65%	93.88%	95.32%	96.04%

**Table 3 sensors-21-02181-t003:** The effect of different window size on the accuracy of the model.

Window Size	192	180	170	160	150
ResNet1D-[1,1,1,1]	88.13%	93.17%	94.24%	96.76%	97.12%
ResNet1D-[1,1,1,1]+	88.49%	93.88%	94.96%	96.4%	96.4%

**Table 4 sensors-21-02181-t004:** The number of gestures collected by each volunteer in each location.

	Location 1 in Room1	Location 2 in Room1	Location3 in Room1	Location 4 in Room2	Location 5 in Room2
Volunteer 1	113	114	116	112	114
Volunteer 2	114	99	114	114	114
Volunteer 3	115	113	115	115	116
Volunteer 4	116	114	115	115	114
Volunteer 5	115	115	114	114	114

**Table 5 sensors-21-02181-t005:** The new data is used to continue training on the existing model, and the recognition accuracy is improved.

Slicing	0:1800	0:1790	1:1791	2:1792	3:1793	4:1794	5:1795	6:1796	7:1797	8:1798	9:1799	10:1800
Room1	76.25%	77.13%	80.94%	81.52%	82.4%	83.58%	85.04%	85.63%	86.22%	86.22%	87.1%	87.98%
Room2	78.17%	79.91%	80.79%	80.79%	81.66%	81.66%	82.53%	82.97%	83.41%	83.84%	83.84%	84.28%

## Data Availability

Not applicable.

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
