# Peer review of "SlideAugment: A Simple Data Processing Method to Enhance Human Activity Recognition Accuracy Based on WiFi"

_sensors, 2021, doi:10.3390/s21062181_

Round 1

Reviewer 1 Report

Its purpose was somehow lost in the text of the work. The authors write: "Please note that we achieve the similar results as presented in the original work. The only difference between the experiment in the original work and the proposed work is that we only use the CSI information and human gesture label. "
What then was the meaning of the work if it reproduces known data? It seems to me that it should be more clearly emphasized: it was - it is.

Reviewer 2 Report

The paper is well organized with proper structure and length. Bibliography is sufficient and well given.

Specifically, the technical terms are explained in detail and the topic of the paper is clear and understandable.

The presented methodology and the results are clearly communicated, with the necessary background for the readers included in the paper.

The review of the state-of-the-art is sufficient. It includes references to other relevant studies that have been previously proposed for the discovery of relations.

The novel contribution of the paper is highlighted, as well.

The conclusion section includes a discussion about the results obtained by this work and the previous works on the analysis of the same or similar data.

Reviewer 3 Report

The interesting direction on human activity recognition based on WiFi is studied. Focusing on the issue coming from limited available data, this work proposes a data augmentation method to solve it. Then, some experiments on benchmarks and real applications are carried out to verify the effectiveness of the proposed method. The writing is well and I am optimistic about it.

One suggestion is that in Abstract and Introduction, the new features of the proposed method can be extracted to replace the steps of the method as contributions.

Author Response

Dear Reviewer,

   Thank you for your affirmation of our research and your careful suggestion. We have revised the Abstract and Introduction according to your advice. The revisions in Introduction are highlighted in revised manuscript.

The new Abstract is as follows :

   Currently, there are various works presented in literature regarding the activity recognition based on WiFi. We observe that the existing data sets do not have enough data. In this work, we present a data augmentation method called window slicing. By slicing on the original data, we get multiple samples for one raw data. As a result, the size of the data set can be increased. On the basis of the experiments performed on the public data sets and our collected data sets, we observe that the proposed method assists in improving the results. It is notable that on the public data set, the action recognition accuracy improves from 88.13% to 97.12%. Similarly, the accuracy is also improved for the data set collected in this work. Although the proposed method is simple, it effectively improves the recognition accuracy. It is a general channel state information (CSI) data augmentation method. In addition, the proposed method has good interpretability.

Reviewer 4 Report

This paper reads well and is generally logically structured in a clear narrative. The paper has a descriptive title with an adequate abstract (while adequate it may be improved with additional content relating to the comparative results) and appropriate keywords.

The general standard of English grammar and syntax is acceptable but it is recommended that the size of the paragraphs be checked as some are too large. Also, I found that the figures are in a number of instances too small and the readability would be improved with enlargement. additionally, the tables require reformatting to an appropriate size matching the template margins (this will be very simple to do).

The authors present their proposed methodology with the results which will in my view be of interest to the intended audience. However, I have comments:

  1. The section headed "Backgrounds" (it should be singular not plural ('Background') as is the case for 'data' (not 'datas') is in reality a review of related research? I am not clear if this section is part of an introduction / related research? There are parts which I think are better placed in an introduction.
      1. The introduction and "Backgrounds" should be revised into two clear sections: (a) Introduction setting out the motivation, overview, contribution, and paper structure, and (2) Related Research setting out a review of related studies considered with a comparative analysis setting out the positive and negative aspects of the methods considered.
    1. I missed a discussion section where the threads introduced in the paper are considered with a comparative analysis comparing the results drawn from other studies. There is a need for a new discussion section.
    2. In all such research there will be open research questions identified along with research questions resolved. The statement regarding related research in the conclusion is completely inadequate and there is a need for a new section where open research questions are introduced, discussed, and proposed directions for research are introduced. 
    3. I note the proposed method uses two wireless routers. In typical implementations such a configuration is not normal and there is a single router. Thinking about the proposed method and the practical application I am not clear how the authors propose to implement their proposed approach in 'real-world' situations. There is a need for a scenario-based illustration to demonstrate potential use-cases and implementation to demonstrate the potential for use in the 'real-world'.

In summary, I found this to be a good paper with an interesting approach which will (in my view) be of interest to the intended audience. However, there is a need for revision to address the points raised in my review. Upon appropriate revision the paper will be a potential candidate for publication.

Reviewer 5 Report

ARIL is not defined.

Work is improved if authors identify research gaps based on a systematic literature review 

Describe better Figure 2. The layout of the two indoor environments. What were the size and room 

More details needed about the implementation of Multi Scale 1D ResNet

References used were not completed

Round 2

Reviewer 1 Report

My comments were taken into account by the authors in the new version

Reviewer 4 Report

I have read and considered the authors response and the revised manuscript. In general I am content with the revisions which address the majority of the points raised in my original review. However, the formatting issues remain (see the figures (too small) and the table (table 5) which fails to meet with the template constraints. I expect the formatting issues to be addressed in the proofing process.